# Tumor Marker B7-H6 Bound to the Coiled Coil Peptide-Polymer Conjugate Enables Targeted Therapy by Activating Human Natural Killer Cells

**DOI:** 10.3390/biomedicines9111597

**Published:** 2021-11-02

**Authors:** Barbora Kalousková, Ondřej Skořepa, Denis Cmunt, Celeste Abreu, Kateřina Krejčová, Jan Bláha, Irena Sieglová, Vlastimil Král, Milan Fábry, Robert Pola, Michal Pechar, Ondřej Vaněk

**Affiliations:** 1Department of Biochemistry, Faculty of Science, Charles University, Hlavova 2030, 12840 Prague, Czech Republic; barbora.kalouskova@natur.cuni.cz (B.K.); ondrej.skorepa@natur.cuni.cz (O.S.); cmuntd@seznam.cz (D.C.); desousac@natur.cuni.cz (C.A.); krejcova.katerina27@gmail.com (K.K.); jahabla@gmail.com (J.B.); 2Institute of Molecular Genetics, Czech Academy of Sciences, Vídeňská 1083, 14220 Prague, Czech Republic; sieglova@img.cas.cz (I.S.); kral@img.cas.cz (V.K.); fabry@img.cas.cz (M.F.); 3Institute of Macromolecular Chemistry, Czech Academy of Sciences, Heyrovského nám. 2, 16206 Prague, Czech Republic; pola@imc.cas.cz (R.P.); pechar@imc.cas.cz (M.P.)

**Keywords:** coiled coil, HPMA polymer, NK cell, NKp30, B7-H6, immunotherapy

## Abstract

Targeted cancer immunotherapy is a promising tool for restoring immune surveillance and eradicating cancer cells. Hydrophilic polymers modified with coiled coil peptide tags can be used as universal carriers designed for cell-specific delivery of such biologically active proteins. Here, we describe the preparation of pHPMA-based copolymer conjugated with immunologically active protein B7-H6 via complementary coiled coil VAALEKE (peptide E) and VAALKEK (peptide K) sequences. Receptor B7-H6 was described as a binding partner of NKp30, and its expression has been proven for various tumor cell lines. The binding of B7-H6 to NKp30 activates NK cells and results in Fas ligand or granzyme-mediated apoptosis of target tumor cells. In this work, we optimized the expression of coiled coil tagged B7-H6, its ability to bind activating receptor NKp30 has been confirmed by isothermal titration calorimetry, and the binding stoichiometry of prepared chimeric biopolymer has been characterized by analytical ultracentrifugation. Furthermore, this coiled coil B7-H6-loaded polymer conjugate activates NK cells *in vitro* and, in combination with coiled coil scFv, enables their targeting towards a model tumor cell line. Prepared chimeric biopolymer represents a promising precursor for targeted cancer immunotherapy by activating the cytotoxic activity of natural killer cells.

## 1. Introduction

The immune system is defending our body not only from external threats (e.g., pathogens, toxins) but also from the harm that may come from the inside (e.g., malignant growth of transformed cells) in a process called immunosurveillance [1]. As healthy cells undergo malignant transformation, the dynamic mutual communication with immune cells turns into immunoediting [2,3,4]. Despite all efforts of numerous lymphocyte populations, this process is not always leading to tumor suppression, but may instead result in tumor escape [3,5,6]. This immune system failure is caused by the tumor’s ability to surround itself with a tumor-suppressing microenvironment [7]. Cancer treatment with immunotherapy strikes tumor barriers and facilitates restoration of immunosurveillance.

The key cellular players in immunotherapy are cytotoxic effector cells, such as T-cells (as well as their subpopulations NKT lymphocytes, Tc-lymphocytes, or γδT-lymphocytes) and NK cells [8]. NK cells tend to be overlooked, as they represent ‘only’ around 12% of the blood lymphocytes [9]. However, they have a unique ability to rapidly identify and eliminate transformed or stressed cells [10]. They participate in tumor immunosurveillance and could be used in targeted immunotherapy [11,12,13,14,15,16]. Therapeutic approaches targeting NK cell activation in cancer treatment are based on native NK cell recognition mechanisms. Alternatively, mAbs can be used for immune checkpoint blockade to prevent NK cell suppression [17]. Moreover, NK cells can be harnessed in various adoptive cell therapies where they are *in vitro* activated, re-educated, or genetically modified with tumor-specific chimeric antigen receptors (CARs) [18,19,20].

Another therapeutic strategy is the mediation of the cellular contact between NK cells and cancer cells. Such interaction could be promoted by mAbs that target tumor markers and at the same time activate NK cells via Fcγ receptor (CD16) in antibody-dependent cell-mediated cytotoxicity (ADCC) [21]. Moreover, this interaction may be enhanced using antiCD16 scFv in fusion with antitumor scFv (targeting, for example, CD19, CD33, CD22, or EpCAM); such a molecule is then called bispecific killer engager (BiKE) [22,23]. The BiKE could be further extended with another recognition domain or with the addition of activating IL-15 (TriKE) [24,25]. This concept could be developed and generalized, such immunoactive therapeutics could also consist of the extracellular domains of NK cell receptors (targeting NK cells activating ligands as markers of tumor cells) [26] or with the NK cells activating ligands (targeting different NK cells activating pathways) [27].

NK cells activating receptors play a critical role in triggering NK cells as anti-cancer therapeutic agents; some of them are dominant in this task, namely NKG2D, 2B4, NKp30, or NKp80 [28]. Receptor NKp30 belongs to the natural cytotoxicity receptor (NCR) family, as well as receptors NKp46 and NKp44 [29]. NKp30 is a type I transmembrane protein consisting of an N-terminal immunoglobulin-like domain, stalk region (15 amino acids), transmembrane helix, and C-terminal cytoplasmic domain enabling signalization through association with CD3ζ chain [30]. Structurally, NKp30 is part of the CD28 family of proteins [31]. Three specific cellular ligands of NKp30 have been identified so far: BAG6, B7-H6, and galectin-3, all of which could be found on the tumor surface, but while BAG6 and B7-H6 activate NK cell cytotoxicity, galectin-3 blocks it completely [31,32,33,34].

Protein B7-H6 is expressed exclusively on the surface of tumor cells, which makes this tumor-induced self-molecule a valuable tumor marker triggering NK cells [31,35]. Like other members of the B7 family, B7-H6 is an immunoglobulin-like type I transmembrane protein. Its extracellular part is composed of a membrane-distal IgV-like domain and a membrane-proximal IgC-like domain [5]. Several crystal structures of NKp30:B7-H6 immunocomplex have been published so far, revealing structural features of this interaction [36,37,38].

The potential of NKp30:B7-H6 interaction in immunotherapy has already been explored. One of the first therapeutic constructs successfully triggering NK cells via ADCC was the NKp30-Fc IgG fusion construct, targeting B7-H6 on prostate cancer cells and resulting in *in vivo* tumor removal in mice [39]. Coating lymphomas or breast cancer cells with B7-H6 and thus mimicking an “induced-self” phenotype for NK cell recognition has also been reported as a therapeutic strategy. To this end, B7-H6 in fusion with single-chain fragment variable (scFv) of the 7D8 mAb, that targets CD20^+^ cells, has been used in *in vitro* cytotoxicity assay alone, in combination with similar ULBP2:7D8 construct or with antiCD20 mAb rituximab, demonstrating the synergy among the two [40] or three [41] various activation pathways. Furthermore, B7-H6:antiHER2-scFv fusion protein can also promote targeted killing of breast cancer cells [42]. B7-H6^+^ tumors could also be targeted with NKp30-based CARs composed of NKp30 extracellular domain and T cell receptor activation motifs [43], or CARs engineered with antiB7-H6 scFv [44,45].

Although strategy with bivalent fusion proteins seems promising, expression of such molecules is limited to a single polypeptide chain, and some targets may be expressed with difficulty (protein stability, expression yield) when fused with the partner. However, recombinant therapeutic proteins must be of exceptional purity and stability, even in blood circulation [46]. One possible strategy to improve both the recombinant protein therapeutic presentation and its stability *in vivo* is the use of polymeric carriers. Polymers tailored for specific applications have various uses in biosciences, ranging from biomaterials used to prepare medical devices and tissue-engineering scaffolds or improved protein crystallization reagents [47] to smart drug-delivery agents optimized for cancer vaccination and immunotherapy [48,49,50,51]. Among the latter, the N-(2-hydroxypropyl)methacrylamide (pHPMA)-based copolymers are especially widely studied and developed for various purposes in biomedical applications, primarily as drug carriers. They are water-soluble, non-toxic, non-immunogenic, and biocompatible. Moreover, due to their intrinsic “stealth properties”, the pHPMA copolymers are invisible to major blood plasma proteins and thus avoid unwanted aggregation [52].

The polymeric carrier must be appropriately modified to deliver biologically active molecules into various parts of the organism. In cancer treatment, these molecules could be small drug compounds or more complex and delicate therapeutic proteins. The choice of a linker is crucial in both cases, but could be more problematic for proteins, as the recombinant expression does not offer the possibility to use the same reactive organic chemistries. One possibility is to use a peptide heptad repeat pattern *(abcdefg)_n_* forming helical and supra-helical structures, i.e., so-called coiled coil motifs [53]. Although there are many naturally occurring coiled coil-based protein domains, these polypeptides could also be synthesized *de novo* with enhanced drug delivery properties [54]. Moreover, pHPMA copolymers could carry more than only a single species cargo, which makes them a versatile tool for biomedical applications, as in the case of targeting BCL-1 cells with pHPMA-(VAALEKE)_4_ bearing doxorubicin and scFv of monoclonal antibody B1 attached via complementary (VAALKEK)_4_ sequence [55].

The major advantage of the coiled coil peptide-bearing polymeric carrier is its versatility—practically any biologically active protein equipped with a suitable coiled coil peptide sequence can be attached to the polymeric carrier. Moreover, the coiled coil anchoring system allows attachment of more than one protein ligand to the polymeric carrier, thus providing bispecific or even multi-specific constructs capable of biorecognition with more than one biological target. To our best knowledge, such a polymer-based bispecific system is described in this work for the first time.

It has been repeatedly demonstrated [56,57,58,59,60,61] that polymer systems with coiled coil peptides enable the use of the so-called pre-targeting strategy. This means that first, a tumor-specific targeting protein equipped with a coiled coil tag is administered to the body, followed by a multivalent polymer containing the complementary coiled coil sequence. Kopecek et al. has shown [59] that careful adjustment of the time interval between the administration of the cancer-specific protein and the polymer significantly improves the efficacy of treatment of non-Hodgkin’s lymphomas based on cross-linking CD20 antigens.

Very recent work has reported [62] that treatment of the malignant lymphomas with a combination of two different targeting ligands (antiCD20 and antiCD38 Fab antibody fragments) induces apoptosis resulting from cross-linking of the corresponding receptors after administration of a multivalent albumin-based cross-linking effector. The combination of the two ligands led to a higher level of apoptosis than applying either one of the two proteins. This result suggests that the combination of two ligands attached to one polymeric carrier might increase the therapeutic potential of similar types of nanomedicines.

In this work, the same complementary (VAALEKE)_4_ and (VAALKEK)_4_ coiled coil motifs forming stable heterodimers were used to prepare protein-polymer complexes targeting cancer cells and activating NK cells simultaneously. Malignant cells could be targeted with monoclonal antibodies recognizing common tumor markers, like carbonic anhydrase IX (CAIX), a sign of cellular hypoxia [63]. scFv fragment of such monoclonal antibody M75 with coiled coil motif has been reported as a targeting molecule for solid hypoxic tumors [64]. Therefore, we have functionalized the pHPMA copolymer not only with the coiled coil M75 scFv but also with a newly prepared coiled coil version of the NK cell-activating ligand B7-H6, thus enabling tumor targeting and direct activation of human NK cells.

The binding of activating ligands on peptide-polymer conjugates represents a promising strategy to directed cancer immunotherapy because the described pHPMA copolymers could deliver more than a single bioactive moiety, enabling activation and targeting at the same time. Moreover, the polymer itself represents a promising precursor for targeted cancer immunotherapy due to possibly improved pharmacokinetics such as reduced catalytic degradation, prolonged blood circulation, and enhanced permeation and retention (EPR) effect that conjugation with polymer is usually associated with [65].

## 2. Materials and Methods

### 2.1. Preparation of Polymer-Peptide Conjugates

Synthesis and characterization of the coiled coil peptide bearing polymer (PolCC) have been previously described [64]. Peptides (VAALEKE)_4_, complementary to the (VAALKEK)_4_ peptides present on the recombinant proteins and forming together a coiled coil structure, were attached to HPMA copolymer via azide-alkyne cycloaddition (“click” chemistry). Reactive propargyl groups (13 mol.%) on the polymeric carrier were modified with the VAALEKE peptide (3.9 mol.%) and biotin (0.86 mol.%), resulting in a 56 kDa PolCC conjugate (Figure 1, left-hand side) with approximately eight coiled coil peptides on each polymer chain (40% *w*/*w*). The newly synthesized conjugate based on HPMA copolymer of increased length (PolCC+, Figure 1, right-hand side) was prepared similarly, except for using RAFT polymerization and copper-free click chemistry—full details of PolCC+ conjugate synthesis and characterization are given in Appendix A. Reactive groups (2.5 mol.%) on the polymeric carrier were modified with the VAALEKE peptide (1.6 mol.%) and biotin (1 mol.%), resulting in a 130 kDa PolCC+ conjugate with approximately ten coiled coil peptides on each polymer chain (26% *w*/*w*). The obtained polymer conjugates were purified by size-exclusion chromatography (SEC).

### 2.2. Cell Lines

HEK293T cells were a kind gift from Prof. Radu A. Aricescu [66]. Cells were adapted for cultivation in suspension in EX-CELL 293 serum-free medium (Sigma, St. Louis, MO, USA) supplemented with 4 mM L-glutamine. Cells were maintained in the square-shaped glass bottles placed on an orbital shaker in a humidified incubator (37 °C, 5% CO_2_) [67].

HT-29 cells were provided by the Institute of Molecular Genetics, Czech Academy of Science, Prague. C33 cell line transfected with CAIX gene (C33_CAIX) or with a control vector containing neomycin resistance gene (C33_Neo) were kindly provided by Dr. Eliška Švastová (Institute of Virology, Slovak Academy of Sciences) [68]. Cells were cultivated in high glucose DMEM medium supplemented with 10% FBS, 100 U/mL penicillin, and 100 μg/mL streptomycin. For splitting, cells were treated with trypsin/EDTA. Both C33 transfectants were maintained under 900 μg/mL of G418 antibiotics. If not mentioned otherwise, all cell-culture media and supplements were purchased from Sigma, St. Louis, MO, USA.

### 2.3. Vector Design

The extracellular domain of NKp30-activating ligand B7-H6 was expressed using pTW5sec plasmid, a derivative of pTT5 plasmid backbone [69,70]. This vector contains secretion leader, the extracellular domain of B7-H6 (residues D25–T244 with stabilizing mutation C212S [37]), coiled coil motif consisting of four VAALKEK sequence repeats, a histidine tag, and WPRE element increasing mRNA stability and protein yield. The expressed coiled coil B7-H6 protein thus contains an additional ITG- amino acid sequence at its N-terminus and a -GT(VAALKEK)_4_H_8_G sequence on its C-terminus.

The extracellular domain of B7-H6 without the coiled coil tag (residues D25–L245 with C212S mutation) and ligand-binding domain of NKp30 (residues L19-E130 representing receptor ectodomain without stalk region) were cloned as described previously [37]. Additionally, vectors for a high-affinity variant of both B7-H6 and B7-H6CC proteins, harboring the recently published S60Y, F82W, L129Y mutations enhancing the affinity towards NKp30 [71], were also prepared.

### 2.4. Protein Expression and Purification

Coiled coil B7-H6 (B7-H6CC), high-affinity coiled coil B7-H6 (haB7-H6CC), high-affinity B7-H6 (haB7-H6), wild-type B7-H6, and NKp30 ligand-binding domain were expressed in suspension-adapted HEK293T cells using a high-density transient transfection protocol and 25 kDa linear polyethyleneimine (PEI; Polysciences, Warrington, PA, USA) as a transfection reagent [69]. On the day of transfection, cells were harvested by centrifugation (5 min, 100× *g*, 25 °C) and diluted with fresh EX-CELL 293 medium (Sigma, St. Louis, MO, USA). Next, DNA (1 µg of DNA per 1 × 10^6^ cells) was diluted into PBS, sterilized using a 0.22 µm syringe filter, and added directly to the cell culture. The medium volume was adjusted to reach a cell density of 20 × 10^6^ cells/mL. Transfection reagent in weight ratio 4:1 (4 µg of PEI per 1 µg of DNA) was added immediately to the high-density cell culture with DNA. The cell suspension was incubated at high density for 2 h on the shaker in the humidified incubator at 37 °C and then diluted with fresh medium to the final production density of 2 × 10^6^ cells/mL. After the dilution, antibiotics (penicillin, streptomycin) and valproic acid (2 mM final concentration) were added.

Cell culture was harvested by centrifugation (30 min, 10,000× *g*, 20 °C) 5–7 days post-transfection. The medium was filtered using 0.22 µm Steritop filters (Merck Millipore, Burlington, MA, USA) and stored at −20 °C for subsequent purification. Proteins were purified by immobilized metal ion affinity chromatography followed by buffer exchange and size-exclusion chromatography. Before affinity purification on a HisTrap Talon column (GE Healthcare, Chicago, IL, USA), the medium was diluted (1:1) with binding buffer (50 mM Na_2_HPO_4_, 300 mM NaCl, 10 mM NaN_3_, pH 7.5). The column was connected to the ÄKTAprime FPLC system (GE Healthcare, Chicago, IL, USA). After loading, the column was washed with binding buffer, and the protein was eluted using the binding buffer with 250 mM imidazole. Buffer exchange into a size-exclusion buffer (10 mM HEPES, 150 mM NaCl, 10 mM NaN_3_, pH 7.5) was done by HiPrep 26/10 desalting column (Cytiva, Marlborough, MA, USA). Protein was concentrated using Amicon Ultra (MWCO 10,000) concentrators at 3900× *g* at 20 °C to reach a suitable sample volume for size-exclusion chromatography (SEC) on the Superdex 200 10/300 column (GE Healthcare, Chicago, IL, USA) with SEC buffer as the mobile phase. Representative analytical SEC elution profiles were acquired with the Superdex Increase 200 10/300 column (GE Healthcare, Chicago, IL, USA). Protein was frozen in liquid nitrogen and stored at −80 °C.

Production of the antiCAIX antibody M75 scFv fragment with (VAALKEK)_4_ coiled coil sequence (scFvCC) was described elsewhere [64]. Briefly, the coiled coil-tagged scFv was expressed in *E. coli* strain BL21(DE3) with translocation of the product into periplasmic space. Protein was then purified separately from the periplasmic and cytosolic cell lysate fractions using a pentahistidine tag on Ni-CAM and ion-exchange chromatography on MonoS columns. The scFvCC purified from the cytosolic fraction showed better purity and was used in all subsequent experiments. The binding activity of the prepared scFvCC towards CAIX was verified by ELISA assay as described previously [64].

### 2.5. Isothermal Titration Calorimetry

Thermodynamic interaction parameters were determined using isothermal titration calorimetry (ITC). All measurements were performed in the SEC buffer. For each run, control heat was determined by a buffer–buffer titration.

The interaction of B7-H6CC, haB7-H6, and haB7-H6CC with NKp30 was measured on MicroCal iTC200 (Malvern Panalytical, Westborough, MA, USA). The measurement of B7-H6CC: NKp30 was initiated by 0.4 µL injection of 182 µM NKp30 into a cell containing 25.0 µM B7-H6CC, followed by 15 injections of 2.3 µL volume with the duration of 4 s with 180 s intervals. Similarly, the measurement of haB7-H6: NKp30 was initiated by 0.4 µL injection of 150 µM NKp30 into a cell containing 15.6 µM haB7-H6, followed by 19 injections of 1.8 µL volume with the duration of 3.6 s with 150 s intervals. The measurement of haB7-H6CC: NKp30 was also initiated by 0.4 µL injection of 124 µM NKp30 into a cell containing 22.4 µM haB7-H6CC, followed by 18 injections of 1.8 µL volume with the duration of 3.6 s with 150 s intervals. The interaction of wild-type B7-H6 with NKp30 was measured on MicroCal PEAQ-ITC (Malvern Panalytical). Initial injection of 0.4 µL NKp30 (179 µM) into a cell containing 24.0 µM B7-H6CC was followed by 25 injections of 1.5 µL volume with the duration of 3 s with 150 s intervals. All measurements were performed at 25 °C while stirring the cell at 750 rpm. The data were evaluated using NITPIC (version 1.2.7, University of Texas Southwestern Medical Center, Dallas, TX, USA; [72]), Sedphat (version 15.2c, by Peter Schuck, National Institute of Health, Bethesda, MD, USA; [73]), and GUSSI software (version 1.4.2, University of Texas Southwestern Medical Center, Dallas, TX, USA; [74]). ITC data were interpreted as A + B <=> AB hetero-association, fitting log(Ka), ΔH, and the incompetent fraction of A (B7-H6; stoichiometry N = 1 − incfA) using the Marquardt–Levenberg algorithm.

### 2.6. Sedimentation Analysis

Sedimentation velocity experiments were conducted on an analytical ultracentrifuge ProteomeLab XL-I (Beckman Coulter, Brea, CA, USA) using an An50-Ti rotor and double sector cells [75]. Proteins and their mixtures at various loading concentrations were spun at 20 °C and 48,000 rpm. One hundred absorbance scans per measurement were recorded at 280 nm using the SEC buffer as a reference. Time intervals between scans for B7-H6CC mixtures with polymer or NKp30 were 7.5 min or 9 min, respectively. PolCC+ was analyzed at 50,000 rpm and 226 nm (400 scans per 2 min), and its mixtures with B7-H6CC at 42,000 rpm and 280 nm (100 scans per 7 min). Buffer density, protein partial specific volumes, and particle dimensions were estimated in Sednterp (www.jphilo.mailway.com, accessed on 28 July 2021). Data were analyzed in Sedfit (version 16.35r, by Peter Schuck, National Institute of Health, Bethesda, MD, USA; [76]) using the continuous sedimentation coefficient distribution c(s) model [77]. Figures were prepared in GUSSI (version 1.4.2, University of Texas Southwestern Medical Center, Dallas, TX, USA; [74]).

### 2.7. NK Cell Activation Assay

NK cells were isolated from the blood samples of healthy donors collected at the transfusion station of the Institute of Hematology and Blood Transfusion (IHBT, Prague, Czech Republic) using the RosetteSep NK cell enrichment cocktail (STEMCELL Technologies, Vancouver, BC, Canada) according to the manufacturer’s protocol. Blood was incubated with polyvalent antibodies cocktail, diluted with PBS, topped on the Ficoll high-density medium, and centrifuged (1200× *g*, 25 °C, 20 min) in SepMate tubes (STEMCELL Technologies). Buffy coat zone depleted from non-NK cell lymphocytes was collected. Isolated cells were washed with PBS, then in 5 mL of red blood cell lysis buffer (eBioscience, San Diego, CA, USA) and PBS again; each washing step was followed by centrifugation (300× *g*, 25 °C, 10 min). Finally, cells were resuspended in RPMI medium (Sigma, St. Louis, MO, USA) supplemented with 20% heat-inactivated FBS, 100 U/mL penicillin, 100 µg/mL streptomycin, 160 ng/mL IL-2 and kept overnight in a humidified incubator (37 °C, 5% CO_2_).

Histidine-tagged proteins were immobilized on the Ni^2+^ chelate-coated 96-well plates (Thermo Fisher Scientific, Waltham, MA, USA). For one well, the total amount of protein (0.04, 0.2, 1, 5, or 50 pmol) was diluted into 50 µL of sterile PBS buffer (cell-culture grade), transferred to the plate, and incubated for one hour on an orbital shaker. An unbound protein was washed away with PBS. NK cells were transferred to the wells in 100 µL of the complete medium. Each well contained the same number of cells in a given experiment, but the total number of cells varied between 1.5–2.5 × 10^5^ in individual experiments depending on the blood donor. NK cells were incubated with the proteins on the plate for 4 h (orbital shaker, humidified incubator, 37 °C, 5% CO_2_).

NK cells were then transferred into a clean U-bottom-shaped 96-well plate, centrifuged (300× *g*, 4 °C, 5 min), and the medium was discarded. Cells were stained for 30 min on ice with antiCD107a antibody conjugated with APC and washed three times with staining buffer (PBS supplemented with 1% FBS, centrifugation 300× *g*, 4 °C, 5 min). After the final wash, cells were resuspended in 120 µL of staining buffer and analyzed by flow cytometer BD LSR II (BD Biosciences, San Jose, CA, USA) using HTS sample mode. Data were analyzed using FlowJo software (version 10.8.0, Becton, Dickinson and Company, Ashland, OR, USA). The Mann–Whitney U test with a two-tailed hypothesis was used to compare the means.

### 2.8. Cell Staining with Polymer–Protein Complexes

First, expression of CAIX on the cell surface of HT-29 cells and C33 transfectants was verified using recombinant M75 primary antibody (kind gift from Vlastimil Král) and goat anti-mouse IgG secondary antibody with AlexaFluor405 fluorescent label (Thermo Fisher Scientific, Waltham, MA, USA) with standard staining protocol.

To perform flow cytometry measurements with direct detection of polymer –protein complexes bound to the cell surface (HT-29 and C33 transfectants), we coupled coiled coil proteins with fluorescent probes using NHS-chemistry; more precisely, scFvCC was labeled with AlexaFluor488 and B7-H6CC with AlexaFluor647 (both labels from Life Technologies, Carlsbad, CA, USA). Cells were detached from culture flasks with accutase solution (Capricorn Scientific, Ebsdorfergrund, Hesse, Germany), washed, and then resuspended in the staining buffer. Cells were labeled for 45 min on ice with polymer–protein particles; the pHPMA polymer concentration in staining solution was 1 µM, the ratio between scFvCC:PolCC:B7-H6CC was 2:1:2, controls such as PolCC alone, scFvCC:PolCC or PolCC:B7H6CC were also included. Cells were washed twice (centrifugation 300× *g*, 4 °C, 5 min) with the staining buffer before fluorescent labeling. Samples were analyzed with the flow cytometer BD LSR II (BD Biosciences, San Jose, CA, USA) and FlowJo software (version 10.8.0, Becton, Dickinson and Company, Ashland, OR, USA).

## 3. Results and Discussion

### 3.1. Design and Production of Coiled Coil B7-H6

The expression construct of NKp30 activating ligand B7-H6 fused with coiled coil motif is based on the soluble B7-H6 construct described before [37] comprising the entire B7-H6 extracellular domain (residues D25–T244) and harboring C212S mutation relieving the molecule of an odd cysteine, a trait known to destabilize recombinant proteins [37,78]. This construct, referred to as wild-type B7-H6 in the following text, was then extended with 28 amino acids (VAALKEK)_4_ coiled coil motif [64] followed by a polyhistidine tag on its C-terminus to create the coiled coil B7-H6 (B7-H6CC; Figure 1a,b). As B7-H6 is a single-pass type I membrane protein [31], this arrangement respects its natural structural features, leaving the N-terminal domain containing an interaction interface with the NKp30 receptor unmodified, while C-terminus, which is in the full-length B7-H6 oriented towards cell membrane, contains the coiled coil sequence. We attempted to produce a B7-H6 construct with the coiled coil motif on its N-terminus, but protein expression was unsuccessful, confirming the need to respect natural receptor arrangement.

The interaction interface between the NKp30 receptor and B7-H6 involves the N-terminal IgV-like domain of B7-H6, which is sufficient for binding [79]. Recently, point mutations increasing B7-H6 binding affinity towards NKp30 have been identified [71]. To include this high-affinity variant of B7-H6 in our study, we chose an S60Y, F82W, L129Y B7-H6 triple mutant reported having the potential to improve NKp30 receptor recognition and thus antitumor properties of B7-H6-based recombinant immunotherapeutics [71].

Although stable cell line generation is required to produce some NK cell receptors or ligands [80], for B7-H6CC, a sufficient yield was achieved by transient expression in the HEK293T cell line using a high-density transfection protocol. The protein was secreted into medium and purified by immobilized metal ion affinity chromatography (IMAC) followed by buffer exchange and size exclusion chromatography (SEC). Immediate buffer exchange on a desalting column was necessary because B7-H6CC protein tends to precipitate in the IMAC elution buffer containing imidazole when being concentrated before SEC. This protocol led to sufficient protein production yields of up to 20 mg per liter of cell culture, which is less than previously reported for the wild-type B7-H6 [37], pointing to a negative impact of the coiled coil tag on the protein stability.

Representative profiles of analytical size-exclusion chromatography and SDS-PAGE gel showing the quality of the prepared B7-H6CC protein are shown in Figure 1c,d, respectively. In both figures, B7-H6CC is compared with the wild-type B7-H6. The expected molecular weight of B7-H6CC is approximately 43 kDa as it possesses six N-glycosylation sites resulting in several glycoforms and blurred bands on the SDS-PAGE gel (Figure 1d). During SEC, B7-H6CC is eluted as a single peak but with a different elution volume than the wild-type B7-H6. This shift could not be explained solely by the increase in protein mass caused by the coiled coil sequence, which itself has only 3 kDa, but it might result from non-covalent dimerization of B7-H6CC driven by the nature of the coiled coil sequence. Indeed, homodimerization of (VAALKEK)_4_ coiled coil motif has been described previously for the scFvCC fragment of the M75 mAb [64]. This interpretation is indirectly supported by the SDS-PAGE (Figure 1d), where the majority of the coiled coil B7-H6 variants migrate as a monomer under non-reducing conditions and directly by the sedimentation velocity analysis discussed below.

### 3.2. Coiled Coil B7-H6 Binds NKp30 Receptor with an Undisturbed Affinity

To verify that the addition of the coiled coil sequence to B7-H6 did not alter the binding affinity towards its cognate receptor NKp30, we performed control isothermal titration calorimetry (ITC) measurements. We determined the binding parameters of the wild-type B7-H6 to NKp30 first (Figure 2a). The obtained parameters (K_D_ = 668 nM, ΔH = −11.5 kcal/mol, ΔS = −10.4 cal/mol∙K, N = 0.88) did not differ significantly from previously published data [36,37,71]. Next, we performed ITC titration for B7-H6CC (Figure 2b). The observed thermodynamic parameters (K_D_ = 787 nM, ΔH = −11.2 kcal/mol, ΔS = −9.6 cal/mol∙K, N = 0.74) match reasonably well those of the wild-type B7-H6.

Similarly, we performed ITC measurements with the high-affinity B7-H6 variant containing mutations S60Y, F82W, L129Y (haB7-H6) (Figure 2c,d), which according to a previous publication, significantly increased the affinity for NKp30 [71]. The authors of this study used biolayer interferometry (BLI) to determine the thermodynamic parameters. As the reference interacting pair for evaluating the affinity-matured IgV domain variants of B7-H6, they used wild-type IgV domain bound to the humanized Fab of cetuximab in an effector-silenced IgG1 SEED backbone. The measured K_D_ was then 409 nM, which is in reasonably good agreement with our ITC measurements. On the other hand, the affinity values of haB7-H6 differ significantly in our measurements. Compared to the published value of 9.06 nM for affinity-matured B7-H6, we observed more than ten times weaker affinity for haB7-H6, i.e., 115 nM, and for haB7-H6CC even more, 259 nM. The discrepancy in the measured values could be because, in the mentioned publication, the affinity-matured B7-H6 domain is bound not to the natural IgC domain, but to the humanized Fab of cetuximab. Even minor changes in the haB7-H6 IgV domain backbone could affect the binding site for NKp30, and thus the two systems are not directly comparable.

At the same time, however, it is worth noting that BLI is a technique that measures the interaction on the surface, similar to surface plasmon resonance (SPR), which has been previously shown to be impractical for measuring binding partners that may oligomerize [37], as the apparent increase in affinity may be due to avidity contribution of oligomers. In this case, NKp30 can oligomerize when glycosylated [37] and if the C-terminal stalk domain is present [81]. However, the BLI in the aforementioned publication was measured with the soluble extracellular part of NKp30 (L19-T138), which contained a non-negligible part of the stalk domain (ten from the total of fifteen amino acids). Considering that there is no available study that would characterize the oligomeric state of the NKp30 variants with truncated stalk domain, we cannot dismiss the possibility that the contribution of avidity biased the measured BLI values.

The thermodynamic parameters measured for the two high-affinity B7-H6 variants interacting with NKp30 in our ITC analysis were K_D_ = 115 nM, ΔH = −12.7 kcal/mol, ΔS = −10.8 cal/mol∙K, N = 0.91 for the haB7-H6, and K_D_ = 259 nM, ΔH = −11.8 kcal/mol, ΔS = −9.5 cal/mol∙K, N = 0.65 for the haB7-H6CC. Noteworthy is the difference in measured stoichiometry for the coiled coil modified variants relative to B7-H6 lacking the coiled coil tag. This shift in stoichiometry might be caused by dimer formation mediated by the coiled coil sequence, as observed previously [64,82] and also confirmed within this study by analytical ultracentrifugation (AUC) analysis (Figure 3a, following page). It is possible that dimerization of B7-H6 partially sterically hinders the access of NKp30 to the binding site on the second B7-H6 unit in the dimer. Such steric barrier would affect the association and dissociation (k_on_/k_off_) of NKp30:B7-H6CC complex, thus explaining lower affinities observed for the B7-H6 coiled coil constructs compared to the untagged proteins.

### 3.3. B7-H6CC Binds to the Polymer-Peptide Conjugate

To analyze the hydrodynamic behavior of B7-H6CC as well as its interaction with the pHPMA polymer modified with the complementary (VAALEKE)_4_ coiled coil peptide (PolCC), we employed sedimentation velocity analysis in an analytical ultracentrifuge (SV-AUC). The B7-H6CC protein alone showed two significant peaks in its sedimentation coefficient distribution (Figure 3a, purple curve) with s_20,w_ values of 2.88 S and 4.17 S corresponding to B7-H6CC monomer and dimer, respectively. The peaks are not resolved well in the distribution, which is typical for a monomer-dimer equilibrium. Wild-type B7-H6 is a monomer in solution [37]; thus, we may attribute the dimer formation to the presence of the coiled coil tag. As mentioned above, we observed a similar effect of the (VAALKEK)_4_ coiled coil tag before in the case of the scFvCC fragment of the mAb M75 [64]. The fitted ratio of the frictional coefficients f/f_0_ of 1.6 corresponds to an elongated particle, suggesting the antiparallel orientation of peptides in the coiled coil dimer.

Next, we analyzed the hydrodynamic states of B7-H6CC with respect to receptor binding. To this end, we titrated B7-H6CC with the soluble ligand-binding domain of the NKp30 receptor and analyzed if the interaction affects the oligomeric state of B7-H6CC or vice versa. As shown in Figure 3a, both monomeric and dimeric B7-H6CC binds NKp30, and this interaction does not disrupt the coiled coil homodimer. Figure 3b shows weight-average S values for the whole sedimentation coefficient distributions plotted against the NKp30 receptor concentration (sw isotherm). The maximum of this curve corresponds to the maximal complex formation observed at 16 μM NKp30 concentration, thus confirming the expected 1:1 overall stoichiometry of this protein-protein interaction. The data could be fitted with the A + B <=> AB binding model, keeping the known values of B7-H6CC and NKp30 sedimentation and extinction coefficients constant while fitting the affinity and sed. coefficient of the resultant complex. In this way, we obtained an average 4.88 S value for the complex and a K_D_ value of 895 nM (N = 0.78), matching reasonably well the ITC results described above (Figure 2b).

As shown previously, the (VAALKEK)_4_ coiled coil homodimer is relatively weak and is easily disrupted in the presence of the complementary (VAALEKE)_4_ peptide bearing opposite charges [64]; therefore, it should not represent any hurdle in the intended use of the B7-H6CC protein. To test if this is also true for B7-H6CC, we mixed PolCC with an increasing molar excess of B7-H6CC and analyzed the resulting polymer–protein complexes (Figure 3c). The equimolar mixture showed a complete absence of the B7-H6CC homodimer (Figure 3c, compare the purple and blue curve), confirming the preferred hetero-association of the used coiled coil peptide pair. Upon increase of the molar ratio between the polymer and the protein, the observed distributions shift from a 1:1 (3.5 S species) to higher stoichiometries, mostly 1:2 (4.5 S), 1:3 (5.6 S), and 1:4 (7.2 S) and only limited number of even larger polymer–protein complexes.

Considering that there are on average eight (VAALEKE)_4_ peptides per PolCC conjugate, these results show that not all these peptides are accessible for B7-H6CC binding. To improve this and create genuinely multivalent polymer–protein complexes, we synthesized a new version of the coiled coil peptide-polymer conjugate (for details, see Appendix A). This novel conjugate, PolCC+, has a similar number of (VAALEKE)_4_ peptides per polymer chain; however, the length of the pHPMA polymer backbone was increased more than twofold, resulting in a lower density of the (VAALEKE)_4_ peptides on the polymer chain and thus potentially fewer sterical clashes between the neighboring bound B7-H6CC molecules. When titrated with B7-H6CC and analyzed by SV-AUC, the new PolCC+ conjugate indeed displayed a considerably higher proportion of larger polymer–protein complexes (Figure 3d).

Finally, we analyzed the interaction of the NKp30 receptor with B7-H6CC when bound to the PolCC (Figure 3e). We prepared the polymer–protein complexes in a 1:3 molar ratio and analyzed them in the presence or absence of NKp30, equimolar to the B7-H6CC. Comparison of the resultant sedimentation coefficient distribution curves clearly shows a significant shift of the observed S values when NKp30 was added, relative to both B7-H6CC:PolCC only (green vs. orange curve) and B7-H6CC:NKp30 only (cyan vs. orange curve). This shift proves the simultaneous complex formation between PolCC, B7-H6CC, and NKp30. Furthermore, NKp30 binds to all species present in the PolCC:B7-H6CC mixture, thus confirming the potential for receptor clustering induced by interaction with such polyvalent polymer–ligand complexes.

### 3.4. B7-H6CC and B7-H6CC:PolCC Complexes Activate NK Cells

The biological activity of the prepared haB7-H6 and haB7-H6CC proteins was tested in the NK cell degranulation assay. Recombinant proteins were immobilized on the Ni^2+^ chelate-coated plates and incubated with human NK cells isolated from the blood samples of healthy donors. Activation was analyzed by flow cytometry, detecting CD107a on the NK cell surface as a degranulation marker (Figure 4). Statistical analysis of the data is given in Appendix A.

After isolation, NK cells were pre-activated by overnight cultivation in the presence of 160 ng/mL IL-2. Recombinant proteins were attached via their polyhistidine tag to the well’s surface in different concentrations, resulting in various activating conditions. The total amount of the protein varied from 0.04 up to 50 pmol, which were chosen as sub-saturating and over-saturating, respectively, according to the manufacturer´s declaration of the wells’ binding capacity. The excess of the protein was washed away with PBS, so unbound NK cell-activating ligand B7-H6 could not block NK cell activation, as described for soluble B7-H6 escaping in the blood circulation due to its proteolytic shedding from the surface of tumor cells [83].

In the first experiment, we tested multiple concentrations of B7-H6 and B7-H6CC in the presence or absence of the coiled coil peptide-polymer conjugate (Figure 4a). In the second experiment, we compared a limited dilution series of B7-H6CC and its high-affinity variant haB7-H6CC to compare the capacity of both proteins to induce NK cell degranulation (Figure 4b).

As presented in Figure 4a, the biological activity of B7-H6CC does not significantly differ (*p* < 0.05) from the activity of non-modified B7-H6 or is even higher. Moreover, the biological activity of the B7-H6CC is not disturbed when bound to the PolCC. We may also conclude that the (VAALKEK)_4_ peptide linker does not interfere with the B7-H6 binding to NKp30, both in solution and on the NK cell surface, and thus is suitable for attachment of the protein to the pHPMA coiled coil peptide-polymer conjugate. Next, we compared the NK cell-activating potential of B7-H6CC with its haB7-H6CC high-affinity version (Figure 4b). The appropriate concentration range to observe the differences was assessed by dilution from 5 pmol to the lower values (1, 0.2, 0.04 pmol), as the differences were better distinguishable in under-saturated conditions. Significant differences (*p* < 0.05) were observed in conditions with 1 or 0.2 pmol of added protein where haB7-H6CC exhibited higher biological activity than B7-H6CC. In this study, we observed significant differences between the response capability of NK cells from different donors. Thus, the NK cell activation level (expressed as a percentage of CD107a positive cells upon stimulation) was always compared only for the single given donor. Data are presented for each donor separately; global trends of donor-averaged data are presented in Appendix A for illustration only. The variation might correlate with the current state of health or intrinsic differences between the donors.

### 3.5. Polymer–Protein Complexes Bind to the Target Tumor Cell Line 

To analyze if the prepared polymer–protein complexes could be targeted to a cancer cell line, we used the previously described scFvCC fragment of the M75 mAb recognizing the carbonic anhydrase IX tumor marker (CAIX) [64]. As CAIX expression is regulated by hypoxia-inducible factor-1 (HIF-1), the amount of gene transcript correlates with the oxygen supply; thus, routinely used cancer cell lines such as HeLa or A549 cultivated in normoxia exhibit no or very low expression levels of CAIX [84,85]. Therefore we used colorectal carcinoma cell line HT-29, CAIX-positive in normoxic conditions, and C33 cell line transfected with the CAIX gene (C33_CAIX) and a control C33 line transfected with the same vector carrying only the same resistance gene (C33_Neo) [68]. Cells were cultivated at normal oxygen levels. The expression of CAIX was verified with M75 antibody resulting in CAIX-high cells (C33_CAIX), CAIX-medium cells (HT-29), and CAIX-low cells (CAIX_Neo) (Figure 5a).

PolCC was mixed with either B7-H6CC alone or with both B7-H6CC and scFvCC at a 1:1 molar ratio simultaneously, and binding of polymer–protein complexes to the CAIX-expressing cell lines was assessed by flow cytometry with direct detection by using an AlexaFluor647-labeled B7-H6CC. As seen from the histogram comparison in Figure 5a, scFvCC:PolCC:B7-H6CC binds specifically on the HT-29 and C33_CAIX cell surface while PolCC:B7-H6CC, lacking the CAIX-targeting scFvCC moiety, shows only a low background signal. The signal intensity of scFvCC:PolCC:B7-H6CC-labeled cells correlates with CAIX expression levels, and in the case of the C33_Neo cell line, only the background signal was observed. This analysis confirms that both coiled coil modified proteins, scFvCC and B7-H6CC, are bound to the complementary coiled coil peptide-polymer conjugate simultaneously.

To directly visualize the simultaneous presence of B7-H6CC and scFvCC on the coiled coil peptide-polymer conjugate, both proteins were fluorescently labeled, scFvCC with AlexaFluor488, and already used B7-H6CC with AlexaFluor647. HT-29 cells were decorated with polymer–protein complexes that always contained the targeting and activating moiety, but different combinations of labeled and non-modified protein variants (Figure 5b). This staining resulted in non-stained cells defining experimental background (Figure 5b, in red); single stained cells in the blue channel, when only scFv_AF488 was detectable (Figure 5b, in light blue); single stained cells in the red channel, when by contrast only B7-H6CC_AF647 was detectable; and finally, the cells labeled in both channels proving that the polymer carries both proteins (Figure 5b, in dark blue).

A minor population of the cells labeled with both colors (using scFv_AF488 + PolCC + B7-H6CC_AF647) and having the signal positive only in the red channel for B7-H6_AF647 could be seen as well. These cells are most likely still labeled with complete polymer–protein complexes, which, however, contain residual unstained scFvCC remaining in the NHS-labelling mixture.

To conclude the binding experiments, scFvCC derived from M75 antibody and modified with (VAALKEK)_4_ coiled coil sequence could be bound to the complementary coiled coil peptide-polymer conjugate. Moreover, it can also target this carrier to the cell surface and bring to the membrane proximity the carrier cargo. Such cargo was in this study NK cell-activating ligand B7-H6, also modified with (VAALKEK)_4_ coiled coil sequence, with undisturbed biological activity to trigger the degranulation of NK cells. These properties make the coiled coil peptide-polymer conjugate suitable for delivering bispecific immunoactive complexes targeting the surface of cancer cells.

## 4. Conclusions

Biocompatible pHPMA copolymers loaded with bioactive protein compounds are a promising tool for targeted cancer immunotherapy. This study describes that such a bioactive protein, NK cell-activating ligand B7-H6, could be recombinantly expressed with a coiled coil motif for polymer attachment. We have also expressed a variant of B7-H6 with a slightly enhanced affinity towards the receptor NKp30. Finally, we have tested the binding of the polymer bearing both targeting moiety (scFvCC targeting carbonic anhydrase IX, a metabolic marker of cancer cells) and the immunoactive moiety (B7-H6CC activating NK cells) to the surface of the cancer cells. Described polymer–protein complexes could be a versatile tool for targeted anti-cancer therapy, as proteins could be easily exchanged or combined to profit from the synergic effect. However, the best stoichiometry, uniform complexes’ composition, and stability must be further investigated before the direct cellular cytotoxicity assays.

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
