# Peer review of "Tumor Marker B7-H6 Bound to the Coiled Coil Peptide-Polymer Conjugate Enables Targeted Therapy by Activating Human Natural Killer Cells"

_biomedicines, 2021, doi:10.3390/biomedicines9111597_

Round 1
Reviewer 1 Report
In this manuscript, the authors explored how copolymer conjugate could be successfully targeted to cells to activate an immune response. A few points/questions are outlined below.
1) The exploration into the utilization of conjugates to activate immune responses is not new to the scientific community. One significant concern with this manuscript is a lack of discussion in the introduction about other techniques and similar approaches with a comparison on how this work is different and improved over previous attempts. The authors need to better clarify the novel of this approach.
2) In Figure 1, the SDS-PAGE, which is used for verification of conjugation, has significant band smears, which is indicative of impurities. Are the authors able to better purify their product to remove contaminants?
3) For someone who is not exceptiaonlly familiar with the methodologies and this field, the results presented do not always show a clear picture. It is suggested that the reviewers find a more "user friendly" way to present data, which would make this study more applicable to a wider audienc.e
4)In figure 4: can the donors be combined and have a data analysis carried out? While maintaining individual donor data is extremely important, visualized average trends would also be impactful
Reviewer 2 Report
The research manuscript entitled “Tumor marker B7-H6 bound to the coiled coil polymeric carrier enables targeted therapy by activating human natural killer cells” by Kalouskova et al. for Biomecines. In this study authors have successfully synthesized the pHPMA-based copolymer and conjugated with coiled coil peptides which was useful for activating to NK cell to target cancer cells. They have also described the receptors of B7-H6 binding partner of NKp30 and corresponding expression to target tumor cell line. This research manuscript is well organized and logically explain the results and provide sufficient data to explain the findings. I will recommend for publication with minor revision.
- Please clarify/redefine the title “coiled coil polymeric carrier”, it seems like authors have used coiled coil polymer, however its not the case. They have used coiled coil peptide to modify the polymer for immunotherapy.
- Introduction section, particularly page 3, lines 113-125 should be edited by new references related to recent engineered polymer nanocarriers in immunotherapy such as Nano Today 2017, 17, 23-37, Vaccines 2021, 9, 935, and Acc Chem Res. 2020, 10, 2094-2105.
- In this study authors have used copolymers combined with different types of peptides; I wonder what is the final formulation structure of polymer-peptide? Please explain in materials method section 2.1.
- Authors are requested to perform the statistical analysis for figure 4 a and b (Both B7-H6 and B7-H6 activate NK cells in the plate activation assay).
